# yQTL Pipeline: A structured computational workflow for large scale quantitative trait loci discovery and downstream visualization

Mengze Li[1,2], Zeyuan Song[3], Anastasia Gurinovich[3,4], Nicholas Schork[5],
Paola Sebastiani[3,4,6], Stefano Monti[1,2,7]*

1 Bioinformatics Program, Faculty of Computing & Data Sciences, Boston University, Boston,
Massachusetts, United States of America, 2 Section of Computational Biomedicine, School of Medicine,
Boston University, Boston, Massachusetts, United States of America, 3 Institute for Clinical Research and
Health Policy Studies, Tufts Medical Center, Boston, Massachusetts, United States of America,
4 Department of Medicine, School of Medicine, Tufts University, Boston, Massachusetts, United States of
America, 5 Quantitative Medicine and Systems Biology, The Translational Genomics Research Institute,
Phoenix, Arizona, United States of America, 6 Data Intensive Study Center, Tufts University, Boston,
Massachusetts, United States of America, 7 Department of Biostatistics, School of Public Health, Boston
University, Boston, Massachusetts, United States of America

* smonti@bu.edu

**Data Availability Statement:** There are ethical restrictions which prohibit the public sharing of minimal data for this study from the New England Centenarians Study. Data are available upon

## Abstract

Quantitative trait loci (QTL) denote regions of DNA whose variation is associated with variations in quantitative traits. QTL discovery is a powerful approach to understand how changes in molecular and clinical phenotypes may be related to DNA sequence changes. However, QTL discovery analysis encompasses multiple analytical steps and the processing of multiple input files, which can be laborious, error prone, and hard to reproduce if performed manually. To facilitate and automate large-scale QTL analysis, we developed the *yQTL Pipeline*, where the '*y*' indicates the dependent quantitative variable being modeled. Prior to the association test, the pipeline supports the calculation or the direct input of pre-defined genome-wide principal components and genetic relationship matrix when applicable. User-specified covariates can also be provided. Depending on whether familial relatedness exists among the subjects, genome-wide association tests will be performed using either a linear mixed-effect model or a linear model. The options to run an ANOVA model or testing the interaction with a covariate are also available. Using the workflow management tool Nextflow, the pipeline parallelizes the analysis steps to optimize run-time and ensure results reproducibility. In addition, a user-friendly R Shiny App is developed to facilitate result visualization. It can generate Manhattan and Miami plots of phenotype traits, genotype-phenotype boxplots, and trait-QTL connection networks. We applied the *yQTL Pipeline* to analyze metabolomics profiles of blood serum from the New England Centenarians Study (NECS) participants. A total of 9.1M SNPs and 1,052 metabolites across 194 participants were analyzed. Using a p-value cutoff 5e-8, we found 14,983 mQTLs associated with 312 metabolites. The built-in parallelization of our pipeline reduced the run time from ~90 min to ~26 min. Visualization using the R Shiny App revealed multiple mQTLs shared across multiple metabolites. The *yQTL Pipeline* is available with documentation on GitHub at https://github.com/montilab/yQTLpipeline.

request from Anna Greenwood, PI of the ELITE Portal DMCC, via email (anna. greenwood@sagebase.org) for researchers who meet the criteria for access to confidential data.

**Funding:** Grant numbers rewarded: U19-AG063893 (PS) UH3-AG064704 (PS, SM) UH3-AG064706 (NS) U19-AG023122 (PS, SM, NS) Funder: National Institute on Aging (NIA). Funder website: https://www.nia.nih.gov/ The funders had no role in study design, data collection and analysis, decision to publish, or preparation of the manuscript.

**Competing interests:** The authors have declared that no competing interests exist.

# 1. Introduction

Genetic association studies aim to test the correlation between disease risks or other phenotypes and genetic variation, with single-nucleotide polymorphisms (SNPs) the most widely used markers of such variation [1, 2]. Quantitative trait loci (QTL) refer to those genetic variations that influence the level of a quantitative trait, for example, expression of a given gene [3].

Several analytical approaches for QTL discovery have been developed to date, examples including *Hail* [4], *MatrixeQTL* [5] and *QTLtools* [6]. However, these tools do not fully account for familial relatedness, which is an essential component in many genetic association studies. *GENESIS* [7] is a package in R that performs genetic association tests while taking into account of familial relatedness, and has been extensively used in GWAS studies [8]. Nevertheless, it can only accommodate one genotype input file and one phenotype at a time, thus its application to QTL discovery becomes inconvenient when faced with a large number of phenotypes and multiple input genotype files.

In addition to the association test, the complete QTL discovery workflow encompasses several preprocessing and post-analysis steps, including conversion of the input genotype file to the correct format, extraction of SNP missingness and frequency information, calculation of genetic principal components (PCs) and genetic relationship matrix (GRM), and merging and visualization of the QTL results. These steps require the execution of multiple commands implemented in different software packages, and can be error prone, time consuming, and difficult to reproduce. We previously developed a Nextflow-based pipeline that incorporates all these steps in a single, reproducible workflow [9]. However, this pipeline is limited to the analysis of one phenotype trait at a time. QTL analysis is often performed over multiple phenotypic traits and processes multiple genotype input files, and visualization of the results can be challenging since the relationship of a large number of genomic loci with multiple traits cannot be easily summarized.

To address these challenges, we developed the *yQTL Pipeline* to incorporate all the analysis steps into a single pipeline. It uses the workflow management tool Nextflow [10] to automate the entire workflow and enables the parallel execution of multiple processes whenever possible.

# 2. Methods: *yQTL Pipeline* design

To ensure modularity, to minimize storage requirements and execution time, and to maximize user control of the analysis steps to be executed, the *yQTL Pipeline* workflow consists of three separate components (shown in Fig 1): *Prepare.nf*, *Analysis.nf*, and *Report.nf*.

*Prepare.nf* performs any data pre-processing when needed, including the conversion of VCF genotype files to GDS format, and obtaining genetic PCs and GRM. Information about the genetic variants, including the allele information, allele frequency and missingness, are also extracted from the genotype data. Next, *Analysis.nf* can be invoked to perform the association test based on the input files either directly provided by the user or from the output of *Prepare. nf*. Finally, *Report.nf* merges the QTL results and generates the plots.

Independent of the analysis workflow, we also make available a stand-alone Nextflow script, *PreQC.nf*, to perform QC on the input genotype data.

This modular design was in part adopted to take full advantage of Nextflow's features. Each Nextflow process first creates a copy of all input files into a "work" directory, which ensures reproducibility, but significantly increases the total execution time as well as the storage requirements, which can become a bottleneck when analyzing large datasets. This is particularly the case in QTL analysis, which takes large genotype input files, executes multiple steps, and generates large-sized result files. Splitting the workflow into three components

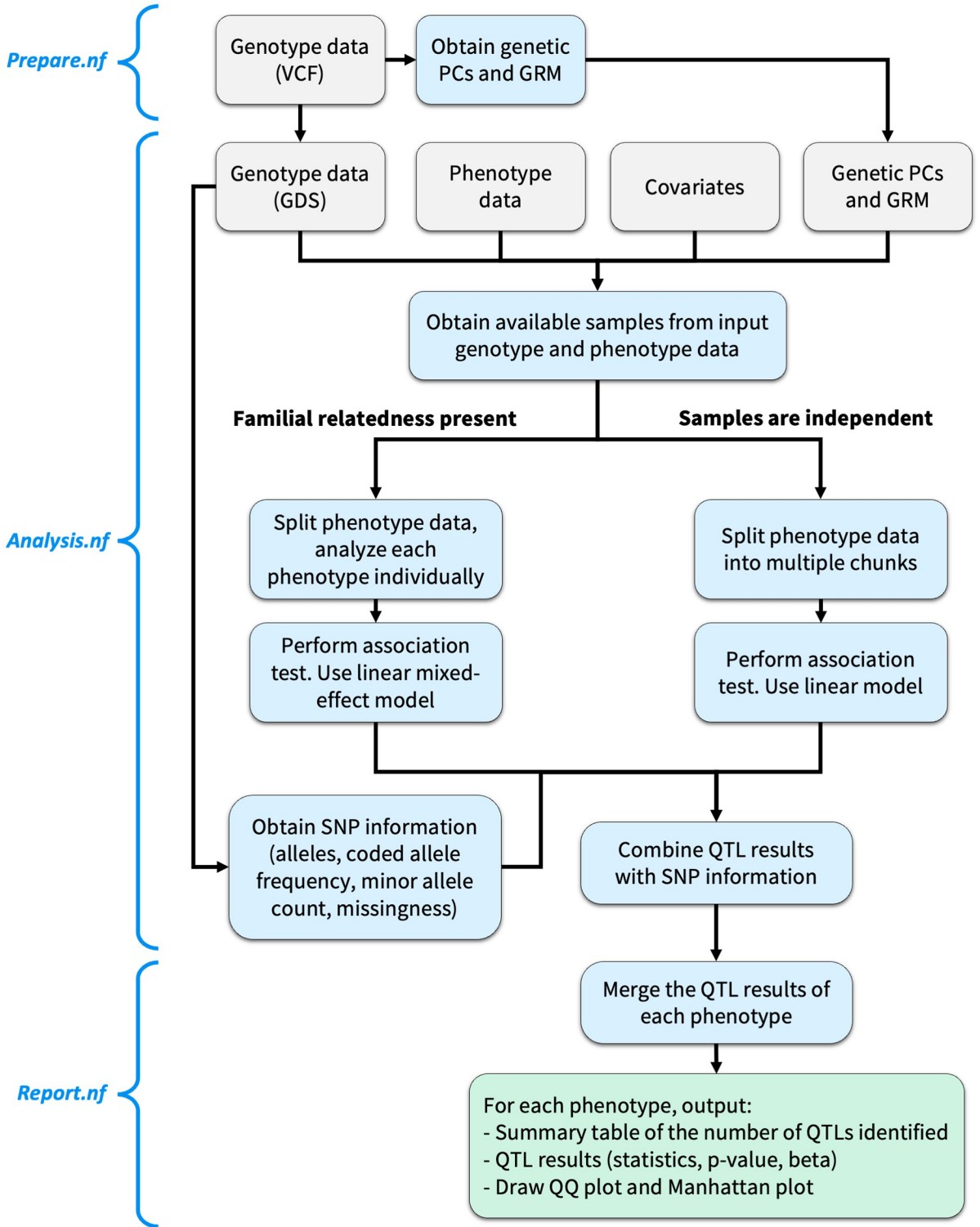

**Fig 1. The *yQTL Pipeline* workflow.** The pipeline is split into three Nextflow steps: *Prepare.nf*, *Analysis.nf*, and *Report.nf*. Two alternative workflows are available for the cases when familial relatedness is present or not. Grey: inputs. Blue: analysis steps and intermediate outputs. Green: final outputs.

significantly reduces the storage and execution footprints, since the input files can be submitted as "values" corresponding to their file paths, rather than actual "files" to be copied.

Throughout the entire pipeline, processes are executed in parallel whenever possible. Parallelization is an essential feature when analyzing a large number of quantitative traits, and/or when the genotype data is provided as multiple files. In large studies, it can translate into hundreds or thousands of independent batch jobs being submitted, which can be executed in parallel and thus highly decrease the run time.

For the configuration and execution of the *yQTL Pipeline*, all that is needed is for the user to specify a configuration file listing the input files and parameters, and to submit three command lines to invoke the entire pipeline. The *yQTL Pipeline* is released under a General Public License 3.0 license. It is publicly available at https://github.com/montilab/yQTLpipeline, including comprehensive documentations of the configuration setup. It supports Linux and OS X operating systems.

## 2.1. Input, configuration, and preparation

The required inputs for the *yQTL Pipeline* include genotype and phenotype data. Optionally, covariates, genetic PCs and GRM can also be included. A more detailed description of each of the input parameters is provided in the GitHub documentation.

**2.1.1. Genotype data.** The pipeline supports either VCF or GDS input format for genotype data. If VCF files are provided, these will be converted to GDS format by running *Prepare.nf*. In addition, the user can specify whether to use the imputed dosage entry or the genotype count entry.

**2.1.2. Phenotype and covariates data.** Phenotype and covariates data should be entered as a data frame in either RDS (R Data Serialization), CSV (comma separated text file) or TXT (tab separated text file) format, with rows denoting samples and columns denoting the phenotypes to identify QTLs from (i.e., the 'y' in the model). There should be a column named "sample.id" to be matched with sample ids in the genetic data files. In addition, the user needs to input a text file that contains all the phenotype trait names to analyze, corresponding to the column names in the phenotype file. The user can specify both numerical and categorical covariates to include.

Genetic PCs, as well as GRM when familial relatedness is presented in the data, can be estimated using different types of computational tools. The *yQTL pipeline* applies PC-AiR [11] and PC-Relate [12] to perform the tasks and is achieved by running *Prepare.nf*. Alternatively, if pre-calculated genetic PCs and GRM are available, they can be provided as RDS-formatted input files.

**2.1.3. An option to analyze a subset of samples and/or SNPs.** By default, the pipeline will perform the analysis using all the samples and all SNPs available in the intersection of all input data files. Alternatively, the analysis can be restricted to a subset of samples and/or a subset of SNPs as specified in user-provided input text files listing the sample and SNP IDs.

**2.1.4. Control Nextflow processes.** Nextflow supports the dispatch of multiple processes in parallel, a feature that can significantly reduce execution time. The user can control the maximum number of processes to run concurrently in the configuration file. When running the pipeline on a high-performance shared computer cluster, the user can also specify distinct resource allocation requirements for each of the pipeline steps in the *SGE* (Sun Grid Engine) configuration file. This is an important feature, as different steps may require drastically different computational resources, and the tailored resource allocation ensures the efficient use of computational (memory and CPU) resources.

**2.1.5. Plotting parameters.** Following the completion of QTL analysis, the *yQTL Pipeline* will generate the Manhattan plots, Miami plots, and QQ (quantile-quantile) plots for each of the phenotypes, as well as a histogram showing the distribution of the phenotype values. The user can specify the minor allele count (MAC) threshold for the SNPs to be included, as well as the resolution and size of the plots. This MAC threshold only affects the plotting and will not filter any of the output QTL results. In addition, genotype-phenotype box plots will be generated for the top SNP in each genotype file input, provided they have passed the user-defined genome-wide significance threshold, which defaults to 5e-8.

## 2.2. QTL analysis workflows

The *yQTL Pipeline* supports two alternative analysis modalities implemented in separate workflows, with the choice to be specified in the parameter "params.pipeline_engine". Available options are "genesis" and "matrixeqtl". The details of each are discussed next.

**2.2.1. Workflow 1: Data with familial relatedness.** When there is known familial relatedness, the user can select *workflow 1* (Fig 1, left side), by setting params.pipeline_engine = "genesis" or "g", which is based on *GENESIS*, and uses a two-step procedure. First, it estimates a "null model" representing the fixed effect of all covariates provided. It then performs association testing for each SNP using a linear mixed-effect model.

*GENESIS* takes a single phenotype, a single genotype file, covariates and a GRM as input. Thus, the pipeline first splits the one multi-phenotype input file into as many single phenotype files, then submits multiple jobs in parallel corresponding to each of the phenotypes and each of the input genotype data files. For instance, if the user wishes to analyze 100 phenotypes and the genotype data is provided as 22 GDS files, corresponding to as many chromosomes, then 2,200 processes will be automatically submitted and run in parallel. The same covariates, PCs and GRM are used across all those processes.

**2.2.2. Workflow 2: Data without familial relatedness.** When the genotype data represent profiles from unrelated samples, the user can opt for *workflow 2* (Fig 1, right side), achieved by setting params.pipeline_engine = "matrixeqtl" or "m", to take advantage of *MatrixeQTL*'s greater efficiency [5]. There are three model options for the user to choose from: "linear", which uses an additive linear model to test for the association of each phenotype with each genetic variant; "category", which model the genotype as categorical variable and runs an ANOVA test; and "interaction", which tests the interaction between the genetic variants and a covariate of choice.

Although there is no set upper limit on how many phenotypes *MatrixeQTL* can handle at once, as the number of phenotypes and the size of the genotype data increase, the required memories increase substantially and may exceed the machine's available resources. To circumvent this problem, the phenotype file will be split into multiple "chunks", with each chunk containing a subset of phenotypes. The user can control the number of phenotypes included in each chunk to balance the memory requirement and total analysis time. The pipeline will then apply *MatrixeQTL* to each phenotype chunk with each genotype input file in parallel. For example, if there are 100 phenotypes, 22 genotype data input files, and a user-specified chunk size of 30 (i.e., 30 phenotypes in each chunk), there would be 4 chunks in total with one chunk containing the last 10 phenotypes, and 88 parallel processes would be submitted. The same covariates are used with all those processes.

## 2.3. Outputs

The intermediate results and the final outputs of the pipeline are saved to separate folders. Log files of all analysis steps are also saved.

1. "1_data" and "1_phenotype_data" (or "1_phenotype_data_chunk") folders contain all data used, including the GDS version of the genotype data if the original inputs were VCF files, and covariate and phenotype data, respectively.

2. "2_SNP_info" folder contains the SNP information, such as allele, missingness and frequency.

3. "3_individual_results" folder contains the QTL results of each phenotype with each genetic data file.

4. "4_ individual_results_SNPinfo" folder is the combination of the two intermediate results above.

5. "5_Results_Summary" folder contains the final output, which includes the merged version of all the QTL results of each of the phenotype including SNP information, a summary table of the number of QTLs identified, as well as the QQ plots, Manhattan plots and Miami plots of each of the phenotype traits. Genotype-phenotype box plots will also be generated if the top SNP in each genotype file input has passed the genome-wide significance threshold provided by the user. Since QTL results are often large data frames, the results are output in RDS format. In addition, the user can setup the configuration file to output QTL results in comma separated text files (CSV format) besides RDS files.

## 2.4. Downstream visualization

We developed an R Shiny App to facilitate post-analysis visualization. In the R Shiny App interface, the user can upload the RDS file generated by the pipeline, or an RDS file in a similar format, i.e., a data frame reporting the phenotype trait names, QTL names, their chromosomal coordinates, and their p-values. The R Shiny App consists of multiple tabs for various visualizations, including the preview of the uploaded QTL result data frame, the Manhattan and Miami plots, the trait-QTL network, and the genotype-phenotype box plot. Fig 2 shows a screenshot of the interface with these different tabs.

The Manhattan plot and Miami plot are two of the most commonly used visualization methods for GWAS analysis since they enable the intuitive identification of significant genetic associations. After uploading the QTL results file, the Manhattan plot and Miami plot of a specific phenotype trait can be generated by selecting a phenotype trait name from the dropdown menu in the R Shiny App interface. In addition, the user can specify a list of SNP IDs in a text input area, separated by comma, to highlight in the plots.

Manhattan and Miami plots can only visualize the results for a single phenotype trait, thus making the comparison across phenotypes difficult. To compare QTL results between multiple phenotype traits, the R Shiny App can also visualize a trait-QTL network. The nodes in the network represent phenotype traits, QTL names (e.g., SNP IDs), and chromosome names. The edges represent significant associations between traits and their QTLs, and top QTLs' co-localization within the same chromosome. The user can specify a p-value threshold, and the trait-QTL network will be generated including only QTLs reaching the threshold. For each phenotype trait, given the large number of adjacent genetic loci in high linkage disequilibrium (LD) with each other, only the most significant genetic locus on each chromosome will be included in the network plot. The resulting network thus displays which phenotype traits have QTLs identified at the selected p-value threshold, which chromosomes those QTLs are in, and whether phenotype traits are sharing (some of) the same QTLs.

In addition, the Shiny App includes the functionality to generate genotype-phenotype box plots. These box plots are particularly useful to assess whether the effect of the alternative

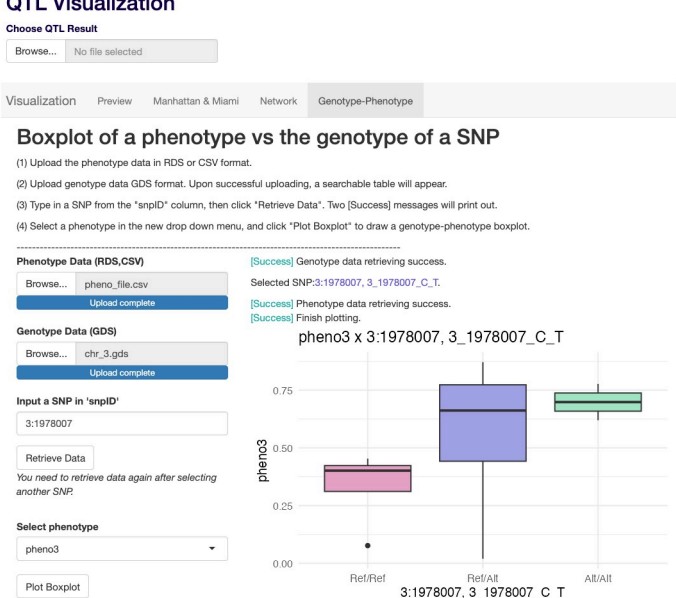

**Fig 2. A screenshot of the R Shiny App.** Tabs for different functionalities can be selected, including the preview table of the uploaded QTL result, the Manhattan and Miami plots of a selected phenotype trait, the trait-QTL network, and the genotype-phenotype boxplot. The screenshot shows the interface for plotting a genotype-phenotype boxplot.

allele is non-additive. Users can upload a genotype file in GDS format and a phenotype file in either CSV or RDS format. Upon selecting a SNP of interest, the app will display a box plot showing the phenotype values for the three genotypes, including homozygous for the reference allele, heterozygous, and homozygous for the alternative allele. The app interface of this functionality is shown in Fig 2.

## 3. Results and discussions: A metabolomics use case of the *yQTL Pipeline*

We illustrate the application of the *yQTL Pipeline* to paired metabolomics and genotype datasets from the New England Centenarians Study (NECS). These datasets profiled 194 NECS participants described in [13]. Age, gender, and years of education were used as covariates. 1,052 metabolites with less than 20% missing values were selected and their expression values were natural log transformed. 9.1M SNPs in the genotype data were used. Since the participants are not genetically related, the pipeline was setup to run with *workflow 2*, in which the linear model implemented in *MatrixeQTL* was applied and samples were considered as independent. The p-value cutoff was set to 1e-3. Since the dataset had previously estimated genetic PCs and the genotype data was already in GDS format, only *Analysis.nf* and *Report.nf* were executed.

Although all 9.1M SNPs were analyzed, to avoid artifacts caused by extremely rare SNPs, only the results from the 3.2M SNPs that have MAC $\geq$ 3 were considered in the following post-GWAS analysis. At the relaxed p-value threshold of 1e-3, all 1,052 metabolites had mQTLs identified. At the genome wide significance threshold (p-value $<$ 5e-8), the list reduced to 312 metabolites. The latter threshold yielded 14,983 mQTLs, including 11,931 unique SNPs, with 3,052 of them being mQTLs shared by at least two metabolites.

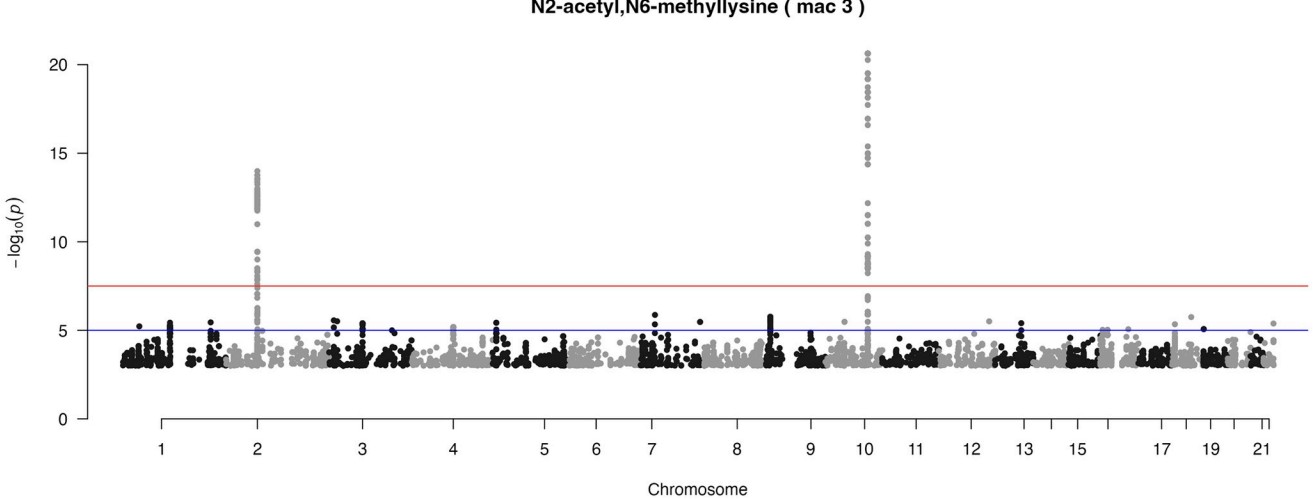

**Fig 3. Example Manhattan plot.** Manhattan plot of N2-acetyl,N6-methyllysine mQTL analysis based on the New England Centenarians Study (NECS) dataset. Minor allele count (MAC) cutoff ≥ 3 was applied to avoid artifacts caused by rare SNPs. Two genome-wide signals on chromosome 2 and chromosome 10 are clearly visible.

Fig 3 shows the Manhattan plot of metabolite N2-acetyl,N6-methyllysine, which is part of the *yQTL Pipeline* output, but can also be generated using the companion R Shiny App. Two genomic loci at chromosomes 2 and 10 were identified at genome-wide significance level (p < 5e-8).

Fig 4 illustrates an example of the trait-QTL network generated by the R Shiny App using the most significant mQTLs obtained (p < 1e-17), which reveals information that would not be easily captured by single phenotype trait visualization methods, such as Manhattan plots. For instance, the network visualization makes it clear that while rs4539242 (bottom of Fig 4) is one of the top QTL associations of N2-acetyl,N6,N6-dimethyllysine, it is also the top QTL of N6-methyllysine. Meanwhile, orotidine (right of Fig 4) has QTLs with p < 1e-17 on both chromosome 14 (top QTL rs192581407) and chromosome 20 (top QTL rs541005701). On the chromosome level, rs768854100 (middle left of Fig 4) on chromosome 10 is the top QTL of undecanedioate, while a few other SNPs on the same chromosome are also the top QTL of other metabolites.

While *MatrixeQTL* has built-in parallelization, the dataset size might preclude the execution at once of the entire analysis. For example, in our use case, it was not possible to run the analysis at once on a machine with 32GB memory. Our pipeline supports the automatic split of the input dataset and the parallel execution of the resulting multiple analyses. If running all analyses sequentially, the total execution time for this use case would have exceeded 90 minutes. Thanks to the parallelization feature of our pipeline, the total run time was reduced to 26 minutes, achieving a ~3.5-fold speed-up. The memory of the compute nodes ranged from 4GB to 32GB, tailored to the requirements of each of the processes. With larger datasets, and when modeling familial relatedness, the execution time reduction would be substantially larger.

## 4. Conclusions

The tools described and results presented provide strong evidence for the usefulness of the *yQTL Pipeline*. By streamlining the analysis process, increasing parallelization, and improving reproducibility of results, and by incorporating multiple steps into rigorously tested and well-

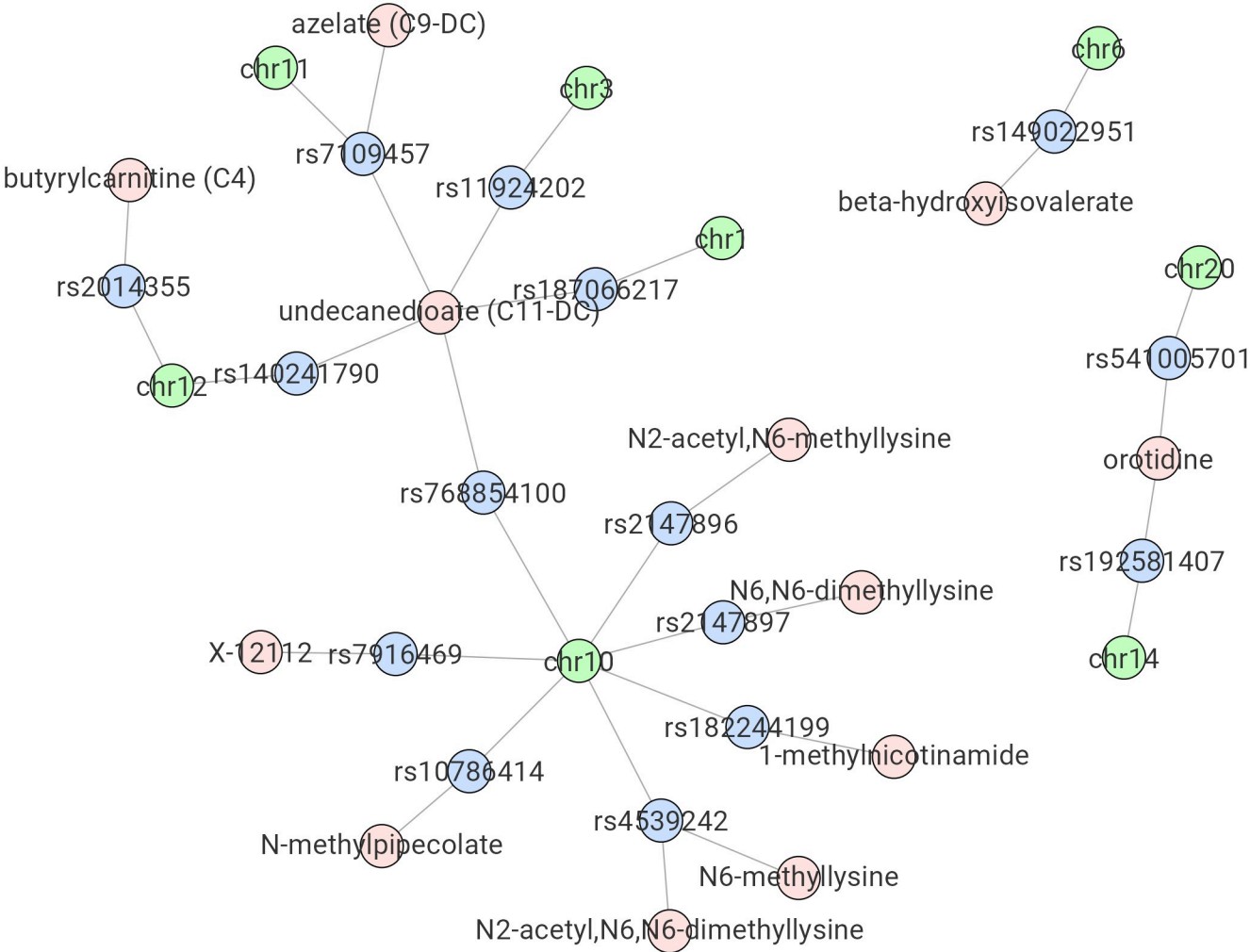

**Fig 4. Example network plot from the R Shiny App.** Results of the New England Centenarians Study (NECS)-based mQTL analysis using a p < 1e-17 threshold are shown. Shared top mQTLs between different metabolites, as well as top mQTLs from different metabolites on the same chromosome are displayed.

documented wrapper workflows, the pipeline will contribute to lowering the barrier to the wide adoption of QTL analysis tools by the research community.

## Acknowledgments

We express our sincere gratitude to our funders and the reviewers who provided insightful comments on the initial draft of this manuscript, contributing to the enhancement of our pipeline's completeness and reliability. Additionally, we express our appreciation to the Massachusetts Green High-Performance Computing Center (MGHPCC), where the analyses were conducted.

## Author Contributions

**Data curation:** Mengze Li.

**Formal analysis:** Mengze Li.

**Funding acquisition:** Nicholas Schork, Paola Sebastiani, Stefano Monti.

**Methodology:** Mengze Li, Zeyuan Song, Anastasia Gurinovich.

**Supervision:** Paola Sebastiani, Stefano Monti.

**Visualization:** Mengze Li.

**Writing – original draft:** Mengze Li.

**Writing – review & editing:** Zeyuan Song, Anastasia Gurinovich, Nicholas Schork, Paola
Sebastiani, Stefano Monti.

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
