## [Decision Letter · Decision Letter 0]

4 Mar 2024

PONE-D-24-03285yQTL Pipeline: a structured computational workflow for large scale quantitative trait loci discovery and downstream visualizationPLOS ONE

Dear Dr. Monti,

Thank you for submitting your manuscript to PLOS ONE. After careful consideration, we feel that it has merit but does not fully meet PLOS ONE’s publication criteria as it currently stands. Therefore, we invite you to submit a revised version of the manuscript that addresses the points raised during the review process.

We look forward to receiving your revised manuscript.

Kind regards,

Chunyu Liu

Academic Editor

PLOS ONE

“We express our sincere gratitude for the funding provided by the National Institute on Aging (NIA), including: U19-AG063893 (PS), UH3-AG064704 (PS, SM), UH3-AG064706 (NS), and U19-AG023122 (PS, SM, NS).”

“Grant numbers rewarded:

U19-AG063893 (PS)

UH3-AG064704 (PS, SM)

UH3-AG064706 (NS)

U19-AG023122 (PS, SM, NS)

Funder: National Institute on Aging (NIA).

Funder website: https://www.nia.nih.gov/

The funder did not participate in the project.”

Reviewers' comments:

Reviewer's Responses to Questions

**Comments to the Author**

1. Is the manuscript technically sound, and do the data support the conclusions?

Reviewer #1: Partly

Reviewer #2: Yes

2. Has the statistical analysis been performed appropriately and rigorously? 

Reviewer #1: N/A

Reviewer #2: Yes

3. Have the authors made all data underlying the findings in their manuscript fully available?

Reviewer #1: No

Reviewer #2: Yes

4. Is the manuscript presented in an intelligible fashion and written in standard English?

Reviewer #1: Yes

Reviewer #2: Yes

5. Review Comments to the Author

Reviewer #1: The study introduces the yQTL Pipeline, a Nextflow-based pipeline facilitating the QTL analysis. This pipeline is designed to handle the multiple, complex steps involved in QTL analysis, from preprocessing input files to running genome-wide association tests and visualizing results. It wraps available QTL tools for both familial and non-familial data. The pipeline incorporates the use of Nextflow for workflow management and parallelization of tasks to optimize run time and ensure reproducibility. It also includes an R Shiny App for user-friendly result visualization. The authors demonstrate the utility of the yQTL Pipeline through its application to analyze metabolomics profiles from the New England Centenarians Study, showing significant speed-up in analysis time and effective visualization of complex QTL data. The English is well written. Though it's generally a useful tool, there is room for improvement before it's accepted for publication.

Comments:

1. Limited Customization: While the pipeline offers some level of customization, users with very specific or unconventional analysis needs may find it less flexible. For example, matrixeQTL provides more models other than linear model, e.g. ANOVA model (modelANOVA) and interaction model (modelLINEAR_CROSS).

2. The authors claimed that the built-in parallelization significantly reduces analysis time, as demonstrated by the case study with a ~3.5-fold speed-up. However, it's not clear how it's improved. MatrixeQTL already has a built-in parallelization mechanism to bin the data and speed it up. Does the author improve the efficiency by making more chunks (thru channels in Nextflow language)?

3. For non-familial data, the pipeline uses matrixeQTL as its engine, "to take advantage of MatrixeQTL’s greater efficiency". However, there are tools such as fastqtl which are known to have better efficiency than MatrixeQTL. The author should consider it and even better make it optional allowing users to choose their own engine.

4. Data visualization: Other than the Manhattan plot, the tool should also generate box plots for the significant phenotype-genotype pairs, which are commonly used in xQTL publications. Also, instead of Manhattan plot, the author could consider using Miami plot by splitting the QTL results with +/- coefficients.

5. Data visualization again: Unlike GWAS result (which is SNP centered), QTL result is for phenotype-genotype pair. So, the author could consider generating two kinds of locus plots, one viewed as the query SNP (where each dot is a gene in the cis window), and one viewed as the query gene (where each dot is a SNP in the cis window).

6. This might be beyond the scope of the work, but since the author developed the yQTL Pipeline to incorporate all the analysis steps into a single pipeline, it's worthy to include pre-QC steps into the pipeline. For example, early work like https://bmcbioinformatics.biomedcentral.com/articles/10.1186/s12859-021-04307-0 has well characterized the necessary step for input data QC for QTL analysis.

7. the link https://github.com/montilab/yQTLpipeline does not work at the moment of review.

Reviewer #2: The yQTL Pipeline presents a significant advancement in the automation of quantitative trait loci (QTL) analysis, transforming a complex and multifaceted process into an accessible, streamlined workflow. This tool capably handles pre-analysis preparations, conducts genome-wide association tests, and facilitates result visualization through an R Shiny App. In its application to the New England Centenarians Study, the pipeline efficiently processed 9.1 million SNPs and 1,052 metabolites across 194 participants, uncovering 14,983 mQTLs while notably decreasing analysis time. The availability of the pipeline and its documentation on GitHub is commendable.

However, several aspects of the study require clarification:

1. The GitHub page mentioned for accessing the pipeline appears to be unavailable or not correctly linked (https://github.com/montilab/yQTLpipeline). Could you provide the correct URL?

2. It would be beneficial to know if the pipeline's runtime has been compared with that of other tools such as GENESIS and/or Matrix eQTL. Can you share any comparative analysis results?

3. Is the pipeline equipped to perform conditional eQTL analysis? If so, could you elaborate on this functionality?

4. Does the pipeline incorporate any genotype quality control (QC) or SNP filtering steps? Clarification on this would be helpful.

5. Regarding the visualization features, is it possible to specifically highlight QTLs of interest for presentation purposes?

These queries are aimed at enhancing the understanding and application of the yQTL Pipeline. I look forward to your responses.

6. PLOS authors have the option to publish the peer review history of their article (what does this mean?). If published, this will include your full peer review and any attached files.

Reviewer #1: **Yes: **Xianjun Dong

Reviewer #2: No

---

## [Author Response · Author response to Decision Letter 0]

12 Apr 2024

See attached document "Review_reply_240404.pdf" (labeled 'Response to Reviewers')

---

## [Decision Letter · Decision Letter 1]

7 May 2024

yQTL Pipeline: a structured computational workflow for large scale quantitative trait loci discovery and downstream visualization

PONE-D-24-03285R1

Dear Dr. Monti,

We’re pleased to inform you that your manuscript has been judged scientifically suitable for publication and will be formally accepted for publication once it meets all outstanding technical requirements.

Kind regards,

Chunyu Liu

Academic Editor

PLOS ONE

Hope the author can address the remaining question raised by reviewer 1 in the final publication.

Reviewers' comments:

Reviewer's Responses to Questions

**Comments to the Author**

1. If the authors have adequately addressed your comments raised in a previous round of review and you feel that this manuscript is now acceptable for publication, you may indicate that here to bypass the “Comments to the Author” section, enter your conflict of interest statement in the “Confidential to Editor” section, and submit your "Accept" recommendation.

Reviewer #1: All comments have been addressed

Reviewer #2: All comments have been addressed

2. Is the manuscript technically sound, and do the data support the conclusions?

Reviewer #1: Yes

Reviewer #2: Yes

3. Has the statistical analysis been performed appropriately and rigorously? 

Reviewer #1: Yes

Reviewer #2: Yes

4. Have the authors made all data underlying the findings in their manuscript fully available?

Reviewer #1: Yes

Reviewer #2: Yes

5. Is the manuscript presented in an intelligible fashion and written in standard English?

Reviewer #1: Yes

Reviewer #2: Yes

6. Review Comments to the Author

Reviewer #1: The authors addressed most of my questions properly, except for the response to Question 3: MatrixeQTL does not take input data format of GDS, but rather same/similar format as fastqtl. So, the response does not answer why they don't use fastqtl as a faster engine.

Reviewer #2: I am pleased to endorse the acceptance of this manuscript for publication. The authors have developed the "yQTL Pipeline," an innovative tool designed to facilitate and automate the discovery of quantitative trait loci (QTL). This work addresses significant challenges in QTL analysis, such as the laborious, error-prone, and often non-reproducible nature of manual processes.

The "yQTL Pipeline" integrates robust computational methodologies, including genome-wide association tests that adjust for familial relatedness and other covariates, and employs the Nextflow tool to parallelize and optimize the analysis process. This results in significant reductions in runtime and enhancements in reproducibility, as demonstrated in their case study of metabolomics profiles from the New England Centenarians Study (NECS). Here, the pipeline efficiently processed over 9.1 million SNPs and 1,052 metabolites, identifying 14,983 mQTLs associated with 312 metabolites, a substantive achievement highlighting the utility of this tool.

This manuscript not only introduces a valuable resource for the genetic research community but also exemplifies the integration of bioinformatics and genomics to advance our understanding of genetic influences on quantitative traits. Therefore, I recommend this manuscript for publication as it provides a significant contribution to the field and a practical tool for researchers.

7. PLOS authors have the option to publish the peer review history of their article (what does this mean?). If published, this will include your full peer review and any attached files.

Reviewer #1: **Yes: **Xianjun Dong

Reviewer #2: No

---

## [Editor Report · Acceptance letter]

24 May 2024

PONE-D-24-03285R1 

PLOS ONE

Dear Dr. Monti, 

I'm pleased to inform you that your manuscript has been deemed suitable for publication in PLOS ONE. Congratulations! Your manuscript is now being handed over to our production team.

Kind regards, 

on behalf of

Dr. Chunyu Liu 

Academic Editor

PLOS ONE